# Aircraft Impact Effects on an Aged NPP

**DOI:** 10.3390/ma14040816

**Published:** 2021-02-08

**Authors:** Rosa Lo Frano

**Affiliations:** Department of Industrial and Civil Engineering (DICI), University of Pisa, 56126 Pisa, Italy; rosa.lofrano@ing.unipi.it

**Keywords:** aircraft impact, ageing, material degradation, FE simulation, safety assessment

## Abstract

The impact of an aircraft is widely known to be one of the worst events that can occur during the operation of a plant (classified for this reason as beyond design). This can become much more catastrophic and lead to the loss of strength of/collapse of the structures when it occurs in the presence of ageing (degradation and alteration) materials. Therefore, since the performance of all plant components may be affected by ageing, there is a need to evaluate the effect that aged components have on system performance and plant safety. This study addresses the numerical simulation of an aged Nuclear Power Plant (NPP) subjected to a military aircraft impact. The effects of impact velocity, direction, and location were investigated together with the more unfavorable conditions to be expected for the plant. The modelling method was also validated based on the results obtained from the experiments of Sugano et al., 1993. Non-linear analyses by means of finite element (FE) MARC code allowed us to simulate the performance of the reinforced concrete containment building and its impact on plant availability and reliability. The results showed that ageing increases a plant’s propensity to suffer damage. The damage at the impact area was confirmed to be dependent on the type of aircraft involved and the target wall thickness. The greater the degradation of the materials, the lower the residual resistance capacity, and the greater the risk of wall perforation.

## 1. Introduction

The need to investigate the effects caused by an aircraft impact on nuclear installation [1,2,3]) is of primary importance and assumes further importance for existing plants that have undergone ageing phenomena or for which an extension of operation (or long-term operation (LTO)) is expected.

Ageing could therefore become an Achilles’ heel for the resistance capacity of structures in such severe accident conditions. As components age, the plant ages too; this means that the cumulative effects of ageing and obsolescence on the safety of nuclear power plants must be re-evaluated periodically to verify that the performances of the single components and the whole plant are within acceptable limits. Therefore, a proper analysis has to be performed to ensure that the effects of ageing will not prevent structures, systems, and components (SSCs) from being able to accomplish their required safety functions. In doing that, according to the International Atomic Energy Agency (IAEA) recommendations [4] the effects of physical ageing, degrading SSC performance, and non-physical ageing (their obsolescence) have to be considered in safety assessments. As for the latter point, it is important to remark that obsolescence occurs also when nuclear equipment becomes out of date in comparison with current codes, requirements, regulations, and technology.

At present, current practice for Nuclear Power Plant (NPP) design against external events provides sufficient safety margin and robustness, allowing the plant to withstand scenarios such as aircraft impact. Nevertheless, an assessment of the plant reliability—in whole or in part—that accounts for ageing degradation is necessary to determine the point in time where profitable operation is no longer possible (IAEA, 2018). This will help defining the optimum preconditions for achieving safe long-term operation (LTO) and maintaining the plant performance level (availability and reliability). A quantification of these combined effects may be evaluated (e.g., structural behavior).

A lot of studies on aircraft impact are available currently in the literature [1,2,3,4,5,6,7,8]; however, none of them investigates the behavior of an aged structure subjected to the impact of an airplane.

The Nuclear Energy Institute (NEI) report [6] provides mainly an overview of the design regulatory requirements; it includes neither detailed analysis procedures nor material models to be adopted for the characterization of structural damage.

Li et al. [7] revisited the relationship between the initial momentum of the aircraft and the total impulse exerted on a rigid target (crushing pressure and dynamic pressure), providing that it is approximately equal to the initial momentum of aircraft. Namely, the initial airplane momentum is transformed into the impact on the target.

Zhang et al. [8] numerically investigated the behavior of the Chinese Ling Ao NPP prestressed containment subjected to an A320 aircraft, establishing that the coupled simulation approach (i.e., missile–target interaction method) may reproduce the impact process (and dynamic response of containment itself) more realistically.

The review study of Jiang and Chorzepa [5] summarizes the characteristics, advantages, and disadvantages of different methodologies (based either on the impact force approach or on interactive impact simulation) that over the last few decades have been used to investigate the aircraft impact. Nevertheless, the discussion here is limited to the nuclear containment. In addition, from the literature it emerges that numerical predictions are difficult to make in terms of analysis methodologies, analysis programs, constitutive models, and element erosion criteria because of the complexity of the phenomena occurring during the impact/crushing (“superposition” of local and global response).

As quoted in [9], the impact loading may be considered a combination of soft (mostly part of aircraft), semi-hard, and hard missile (i.e., engines and landing gear) impacts, while the impact force is strongly dependent on the missile’s deformability. A hard impact leads to impactive loading (energy driven), while a soft impact leads to impulsive load (impulse driven). 

In this framework, further concerns are due to the concrete containment (CC) ageing over time: the greater the ageing, the lesser the load-bearing capacity. Therefore, the important open question is if the actual CC wall thickness is capable of withstanding an aircraft impact.

The following section provides: the qualification of the numerical FEM code used for the numerical investigation (which was conducted based on the experimental results from the full-scale aircraft impact tests performed by Sugano et al. [10,11]);the description of the implemented model and assumptions made;the obtained results. The method to approach aircraft impact is described at length in [2].

## 2. Modelling Qualification 

The study of the penetration and perforation of a thick wall (plate) covers a wide range of problems, interests, and applications. In most problems, the main point of interest is either the damage behind the thickened or reinforced wall or the damage confined to the wall itself. 

According to Backman and Goldsmith [12], the wall of a containment system may be classified as thick depending on whether its rear surface influences or has a significant influence on the penetration process [13]. 

The impact phenomenon is difficult to characterize from a physical point of view because it depends on several factors, such as the type of impact (i.e., soft impact) [14], missile geometry, properties, and material behavior (deformable or hard). Moreover, based primarily upon the ratio of pressures to the projectile strength, it is possible to identify three different impact regimes. In the first regime, termed the low-velocity regime, the impact pressures are so low that the projectile is often assumed to be elastic or rigid. By neglecting the projectile, impact problems are indeed greatly simplified. This regime has been studied extensively; a good number of these studies are summarized in Table 1 of Jiang and Chorzepa [5]. In the second regime, the high-velocity regime, the impact pressures are great enough to cause the extensive deformation or flow of the projectile. 

An analytical model must incorporate erosion and breakup into its solutions. In the third regime, the hypervelocity regime, the impact pressures are much greater than both the involved projectile and the target strength, so erosion dominates the penetration process.

In the present paper, we deal with a soft impact, whose interest resides in the possibility to simulate the interacting structures, such as deformable (to a small or large extent). A soft impact has no rebound and the two mass points stick together. 

The interacting structures become one single body after the crushing and the kinetic energy (depending on the “missile’s” velocity and mass) can be dissipated [14]. This assumption derives from what was shown by the full-scale experiments performed by Sugano et al., in collaboration with the Sandia National Laboratories [10,11,15] and Muto et al. [16]; no rebound of the aircraft occurred.

The qualification of the FEM code [17] was performed by simulating a full-scale aircraft impact test, in which an F4-Phantom of 19 tons weight or its engine travelling at 215 m/s impacted a rigid wall that was 160 mm thick. The size of the impact area is specified as 7 m^2^. Figure 1 shows the impact test (from [15]) and the numerical modelling (Figure 1b) that have been set up and implemented, respectively.

In the qualification analyses by MARC© code using the same initial and boundary conditions, the same geometrical dimensions and material properties have been adopted. The impact duration was 2.1 ms. The striking was simulated by assuming a contact method (touching type); the contact detection between all contact pairs was based on a double-sided method. Moreover, a piecewise linear approximation which was based on the relative displacement between the bodies in contact was also implemented.

### Analysis of the Results

Figure 2 shows the progression of the damage as the airplane penetrates the target wall and the damage caused by the impact force [15,18]: from the top left snapshot to the bottom right one, the impact, penetration, spalling, and crater formed are visible.

Since the target wall was penetrated, the reinforcing rebars resulted uncovered (as is visible in Figure 2). Indeed, no perforation occurred. The simulation results are given in Figure 3 and Figure 4. They represent the evolution of damage from penetration until fragments are projected from the target surface.

Various observations can be made from those figures. At first, material is ejected from the front face of the target as the airplane embeds itself. Not all the material ejected from the front face belongs to the target material, as some is the projectile, which is stripped away. Because the striking airplane is longer than the target is thick, this implies that considerable flattening has occurred.

Figure 3 illustrates that thick target penetration is not one mechanism event; it goes through a series of stages, starting with the embedding of the nose of the airplane into the target and ending with the spalling and/or scabbing of the target material.

Fragmentation finally occurs because of the large amount of energy transferred to the target in short time: the local material.

The calculated mean penetration depth, of about 0.055 m (see Figure 3), is quite similar to the measured experimental value; this confirms to a certain extent the agreement between the experimental and numerical results. Moreover, it is possible to observe the same damaging modes, specifically penetration, with the formation of a narrow cone accompanying cracking (also in the radial direction), and spalling, with the projection of fragments from the proximal face of the target (Figure 3b). From the simulation, we also derived other information, such as the stress distribution, which is not known from the experiments. Figure 5 shows the impact stress distributed over the rear surface: even if in some points it reaches the allowed limit, it did not determine any scabbing.

## 3. Aircraft Impact Assessment: Modelling and Simulation

Because of the safety concerns, we have to prove by means of safety assessment and calculations and build confidence in the safety of the CC and show that the consequences of an aircraft impact on the environment and humans are negligible. These consequences are traditionally categorized into local and global effects. Accordingly, we must distinguish between global and local damage (or local and global response). The global effects include overall axial, bending, and shear effects in the structural elements between the impact area and support locations.

The global damage identifies the collapse of large portions of the building walls, load carrying members, etc., apart from vibration, that can potentially shake the whole building structure. The extent of such a “global failure” will depend primarily on the dynamic characteristics of the CC and missile, and the relative damage is, in general, associated with the excessive deformation of the entire structural system, assuming that local perforation does not occur. The behavior of concrete in compression and its compressive strength become very important in assessing the local behavior, whereas the tensile strength properties and the stress–strain response of reinforcement steel, in many cases, dominate the deflection and vibration behavior of the reinforced concrete structural members.

As for the local effects concerned, according to [19] these are due to hard (or semi-hard) missile impacts, resulting in punching failure mode. The sequence of localized loading effects consists of (Figure 6): (a) penetration into the target; (b) cone cracking and plugging due to the inertial stress wave; (c) spalling with the ejection of target material from the proximal face of the target; (d) radial cracking behind initial wave front; (e) scabbing with the ejection of fragments from the distal face of the target; (f) potentially, perforation completely through the target; and failure. Although these would not, in general, determine a structural collapse, they have to be considered because of the secondary effects on the safety-related systems or components.

In the light of the aforementioned points, the containment structure must have a minimum wall thickness to prevent perforation, even in the presence of penetration and scabbing [20], and fire. This latter would impair the core cooling, as indicated in 10 CFR 50.150 and NEI 07-13, 2011 [6].

For the purposes of this study, a deterministic method which uses dynamic simulations to verify “the acceptance criteria” are respected, as required by the applicants, has been adopted.

The numerical simulation of a reinforced concrete structure is a challenging task due to the inhomogeneity of the material (non-linear behavior) which is also dependent on the strain rate. In order to correctly consider the dynamic characteristics of an impacted structure, a missile–target interaction method with explicit integration techniques has been adopted.

### 3.1. CC Model

The reinforced concrete containment of this study is that of a typical pressurized water reactor (60 m tall, about 45 m diameter and 1 m thickness) resting on a rock soil (“rigid foundation”), represented by means of clamped restraints. To comply with the ASCE rules and ACI standards, the CC reinforcement is made of #8 reinforcing steel bars in the dome, distributed about 30 mm from the center of each direction and each face, #18 vertical bars in the cylindrical part at about 30 mm spacing for each face, and #18 horizontal bars on both sides of the vertical reinforcing grid. The basement has instead steel plates of 13 mm thickness [21]. The material properties of reinforcement are provided in Table 1.

Figure 7 shows the FE model of CC that was made of more than 68,000 solid elements. The concrete structure and the main body of F4 plane are modelled with SOLID-3D Lagrange elements and the internal structures using 3D thick shell elements. The TRUSS-3D elements steel is used for the reinforcement (rebar).

To achieve sufficient accuracy of the dynamic analysis results, sensitivity studies were performed to identify the allowable maximum element size. In determining this, the frequency of the highest mode and the longitudinal propagation velocity of the concrete material were considered. The mesh size was chosen to be uniform in each 15° angular sector of CC discretization, with 20 elements through the thickness (element vertical dimension of 1 m) and about 72 elements across the airplane cross-section.

The connection of the reinforcement and concrete may be a crucial issue. All the models presented in this paper utilize a perfect bond with fully kinematic coupling, therefore the effects of bond slipping cannot be taken into account.

Adequate initial and boundary conditions (continuum mechanics) have been assumed so as to preserve energy and momentum.

The mathematical model is able to describe the characteristic of contact deformation during crushing. The contact force is automatically calculated by a code based on the conservation of the impulse. Moreover, assuming a continuous material, contact theory and structure mechanics allowed us to describe the CC mechanical behavior through its mechanical properties such as Young’s modulus, yielding stress value, and contact type, etc. A proper contact table describing the interaction between the two bodies—i.e., missile and target—was also defined. The contact detection method was described by a stress-based contact algorithm (which allows us to decide whether or not bodies in contact should separate).

In simulations dealing with impact loading, pronounced material nonlinearities occur. The constitutive laws, especially for concrete, must be able to reproduce the relevant phenomena.

The behavior of concrete is assumed to be linear elastic within an elastic limit surface and up to the limit stress. With the onset of plastic strain and its further increase, material damage will rise until failure occurs. This is considered by damage parameters representing the progressive cracking and crushing of concrete caused by the abrupt stiffness changes (due to the propagation of the aircraft missile’s kinetic energy), which scale the failure surface down to a residual strength surface.

The main challenge in non-linear analyses of concrete structures is due to the relatively low tensile strength. The tensile stress causing cracking in concrete is about one tenth of its crushing strength; in reinforced concrete structures, this “weakness” is compensated by the use of reinforcement. In addition, it should be considered that the concrete material strength may increase at elevated strain rates. To account for this effect, which for concrete is particularly due to the inertial effects, a dynamic increase factor (DIF) equal to 1 is considered for steel and concrete, as the dynamic load factor associated with the impactive or impulsive loading is less than 1.2 [22].

The implemented concrete ‘plasticity’ model accounts for material degradation in compression and in tension as impact progresses. Damage effect types are associated with cracking properties, such as critical stress, softening modulus, crushing strain, etc. It is necessary to remark that in the analyses, the compressive stiffness was recovered upon crack closure as the load changed from tension to compression, but the tensile stiffness was not recovered when the load changed from compression to tension. Equations (1) and (2) provide the stress–strain relationships for uniaxial tension and compression:(1)$$\sigma_{y}=\left(1-d_{t}\right)E_{0}\left(\varepsilon_{t}-{\bar{\varepsilon }}_{t}^{p}\right),$$
(2)$$\sigma_{c}=\left(1-d_{tc}\right)E_{0}\left(\varepsilon_{c}-{\bar{\varepsilon }}_{c}^{p}\right),$$
where *d_t_* and *d_tc_* are the tensile and compression damage parameters, respectively. *E*_0_ is the undamaged modulus of elasticity. *ε_t_* and *ε_c_* are the tensile and compression strains, whereas ${\bar{\varepsilon }}_{t}^{p}$ and ${\bar{\varepsilon }}_{c}^{p}$ are the equivalent plastic strains in tension and compression.

The airplane and reinforcement steel non-linear material behavior can be modelled using an elastic-plastic material model with the von Mises yield criteria and the Cowper–Symonds strain rate method. To simulate extreme loading conditions, such as impact phenomena, material models should be strain rate-dependent. Moreover, the dynamic yield strength is considered in the Cowper–Symonds formula:(3)$$\sigma_{y}=\sigma_{ys}\left[1+{\left(\frac{\dot{\varepsilon }}{D}\right)}^{\frac{1}{q}}\right].$$

*σ_y_* and *σ_ys_* are the static and the dynamic yield stress, respectively; $\dot{\varepsilon }$ is the strain rate; D and q are the steel material parameters, equal to 40 1/s and 5, respectively, and are taken from the literature [23]. Further restrictions and simplification have been also imposed on the airplane velocity (assumed constant), impact direction, and internal components.

In preliminarily analyzing the ultimate structural capacity, varying material properties with age (realistic values) are taken into account. Such a variation/degradation is considered in the assessment assuming that the values of the mechanical and physical properties of both the concrete and steel—e.g., yielding strength and Young’s modulus—are equal to 80% of the nominal value.

For the purposes of the simulation, a horizontal impact onto the CC dome is assumed in order to consider the worst accident scenario (Figure 8).

The aircraft crashing was simulated by performing a nonlinear transient analysis and assuming the updated Lagrangian procedure with follower force/stiffness command. For the current analysis, the large strain plasticity procedure is activated. The “large displacement” model option, which causes distributed loads to be based on the current deformed geometry, contributes to a stiffness effect on the tangential stiffness matrix. Erosion is not thus implemented because the model procedure allows us to account for the failure process and handle the large distortion problem (large deformation), such that the actual strength of the structure is calculated by the code by immediately deactivating the damaged elements.

The time step size was set to be equal to 10^−5^ s to properly capture the damage of the material as the impact progresses.

### 3.2. Results Discussion

Because of the strike against the target, a part of the aircraft is crushed while the remaining portion undergoes elastic deformation. The kinetic energy transferred during the impact is, thus, partially dissipated by the crushing and buckling of the plane and partially through the deformation of the walls. The impact force at the time of the crushing of the fuselage (30 ms) is about 89 MN; this value is in good agreement with the Riera impact load (represented with a dashed line in the trends of Figure 9) and with the Reisemann experimental data [18,19,20,21,22,23,24].

Upon impact, a compressive wave propagates away from the impact point; the radial stress built up is tensile in nature. When the CC material stress is greater than the ultimate dynamic strength, radial and/or circumferential cracks occur. As the compressive wave propagates outwards, the hoop or circumferential stresses generate radial cracks due to the Poisson effect.

The containment structure deforms beyond the point of permanent yielding; pronounced damage on the front and rear face occurs as a consequence of the impact.

The extensive penetration of the wall (critical condition) is reached in less time for an aged containment than for an unaged one due to the reduced strength of the materials. Figure 10 provides the contour bands of the stress; with the same impact force transmitted to the structure, the resistant capacity of the containment reduces by about 30%. Furthermore as the results of the aircraft impact, a narrow cone forms at the entry location, confirming the wall penetration (Figure 11). The penetration depth ranges between about 5.8 and 7.7 cm for the unaged and aged structures, respectively. In both the two conditions, the penetration is accompanied by cone cracking and some spalling of the concrete wall, while acceleration propagates faster through the structure. Even if perforation is precluded, the concentrated loading can produce significant scabbing, weakening a portion of the structure up to ten times the missile diameter in each direction and affecting the global behavior.

Figure 12 shows the acceleration propagation inside the CC structure. It can be stated that the observed dynamic response due to the aircraft impact is characterized by much higher acceleration values, especially in the high frequency range (approximately from 25 to 80 Hz), in comparison to further external event loading cases. The high accelerations at the outer CC wall in the vicinity of the impact location are transferred directly to the inner structures.

If the inner and outer structures are separated, the travelling shock waves are attenuated significantly and filtered during propagation all the way down to the foundation and, from there, to the supporting point of the equipment in the inner structure, as shown by the A_3int_ and A_2int_ trends.

As for vibration, the calculated acceleration of about 40 g is not sufficient to damage the internal structures, and particularly the containment vessel.

Finally, it is possible to say that, despite the 20% reduction in the materials’ properties, the strength capacity/integrity of the CC seemed not to be compromised. In fact, the phenomena of instability induced by overturning moments do not appear, nor does the strain exceed the limit value of 0.005, which is the acceptance criteria suggested by the IAEA for the compressive strain of concrete.

## 4. Summary

Prior to investigating the effects of aircraft impact in an existing Gen II PWR containment, the qualification of FEM code, which was used for the simulations, was conducted on the basis of the experimental results available in the literature (a very good agreement was found).

The results showed that ageing increases a plant’s propensity to suffer extensive or localized penetration that was about 5.8 and 7.7 cm for unaged and aged structures, respectively. Moreover, despite a partial wall penetration (a narrow cone was clearly observed), no wall perforation occurred.

The greater the materials’ degradation, the lower the residual resistance capacity and the greater the risk of wall perforation. By assuming the same impact force, the strength capacity of the aged containment is reduced by about 30%.

Generally, the local effects (penetration, spalling, and high-frequency acceleration) do not pose a threat to the safety-related systems necessary to shut down a nuclear plant. Global stability seems to be guaranteed, as overturning moments do not appear, nor do the strains of concrete exceed 0.005 (acceptable criteria value).

## Figures and Tables

**Figure 1 materials-14-00816-f001:**
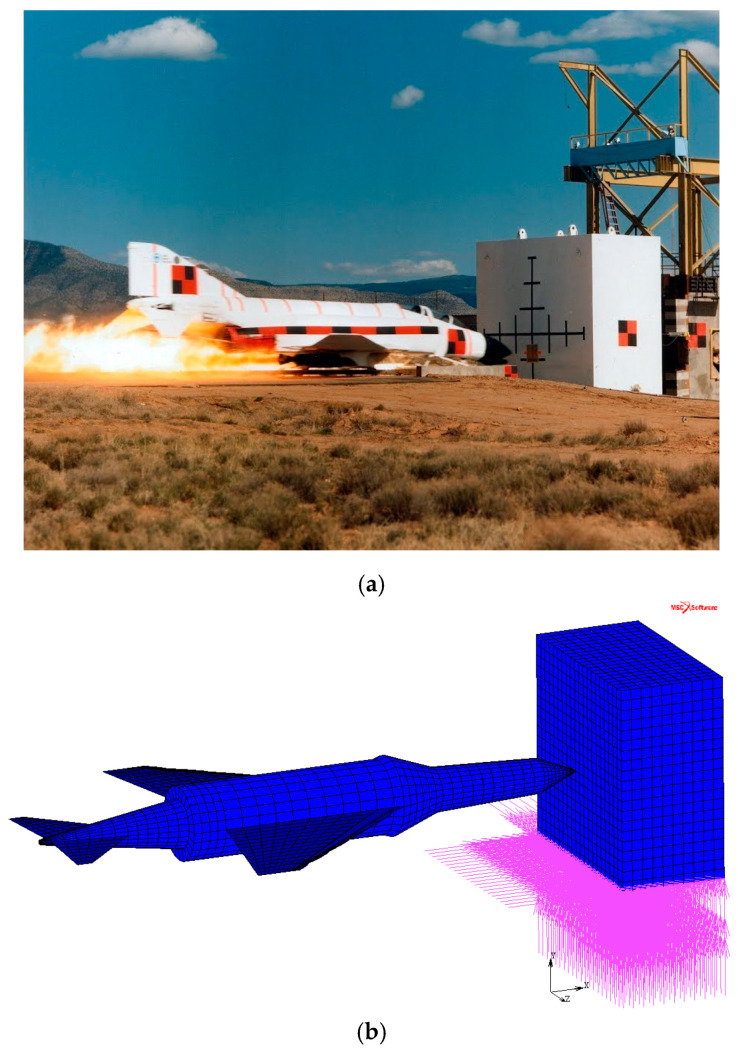
Full-scale aircraft-impact test (**a**) and numerical model (**b**).

**Figure 2 materials-14-00816-f002:**
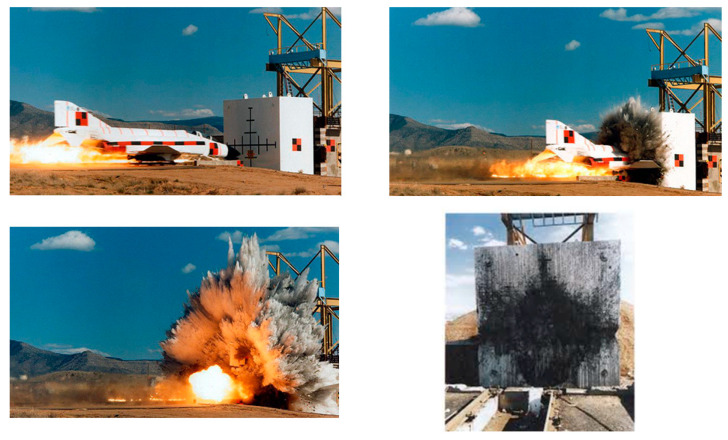
Snapshots from the Phantom F-4 impact test.

**Figure 3 materials-14-00816-f003:**
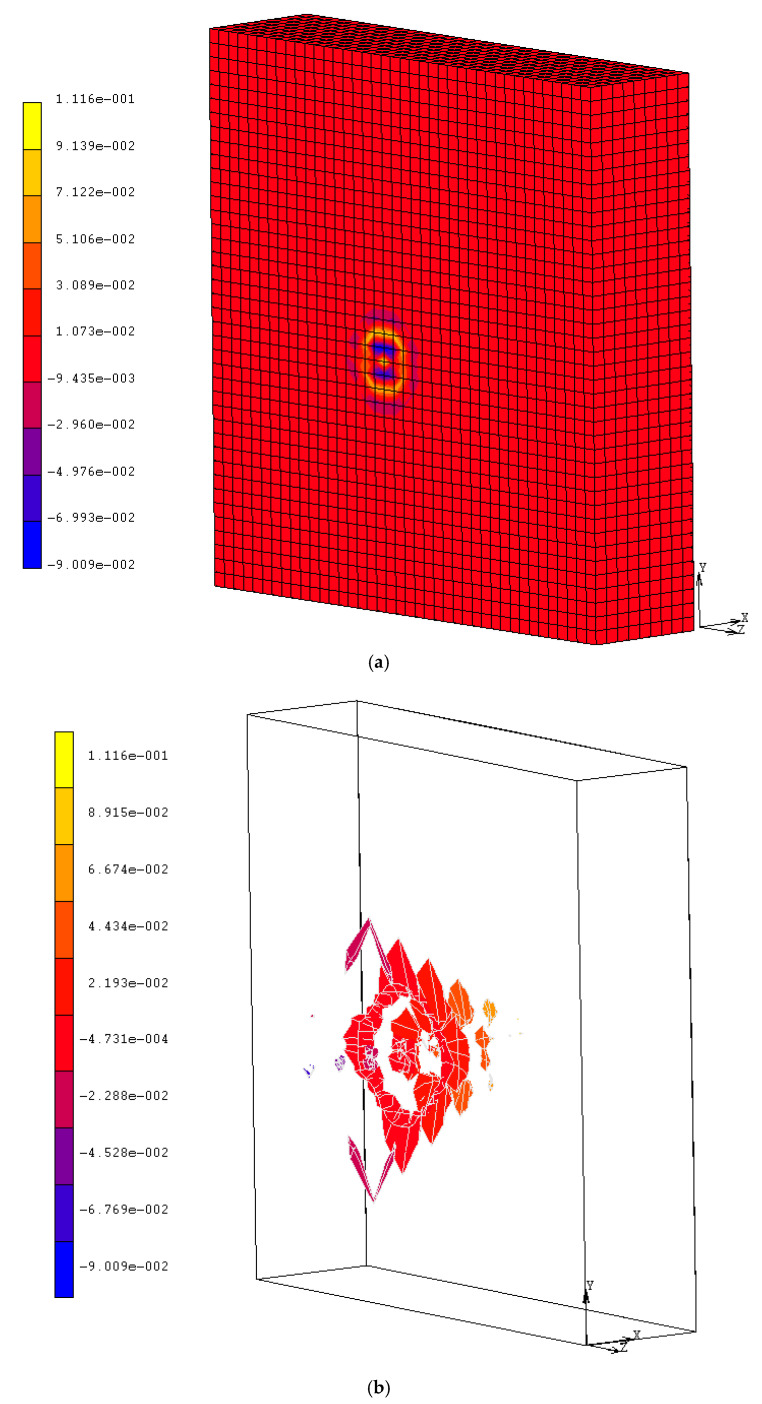
Penetration depth (**a**) of about 10 cm and fragment projection (**b**) at t = 2.1 × 10^−3^ s.

**Figure 4 materials-14-00816-f004:**
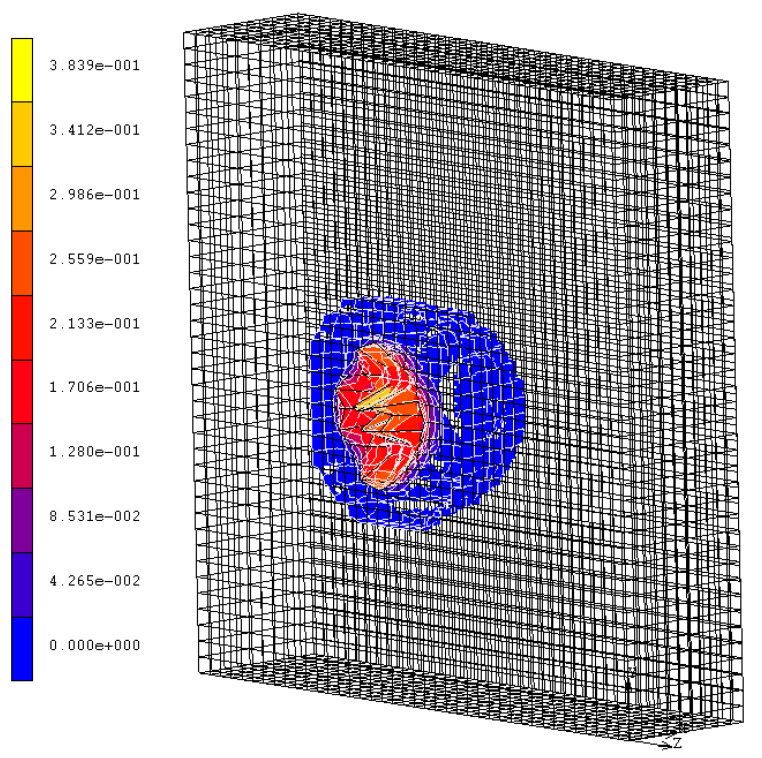
Plastic strain (−) at the front surface impacted: iso-surface representation at t = 2.1 × 10^−3^ s.

**Figure 5 materials-14-00816-f005:**
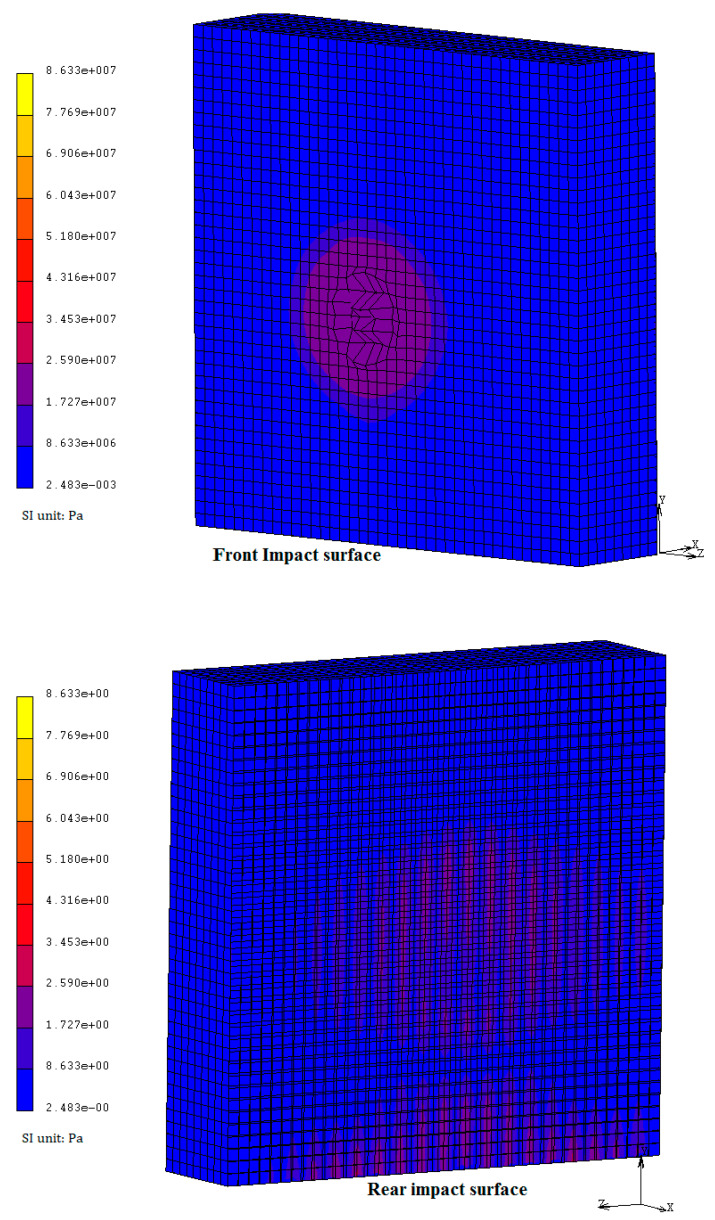
Von Mises stress distribution at 2.1 ms from the impact.

**Figure 6 materials-14-00816-f006:**
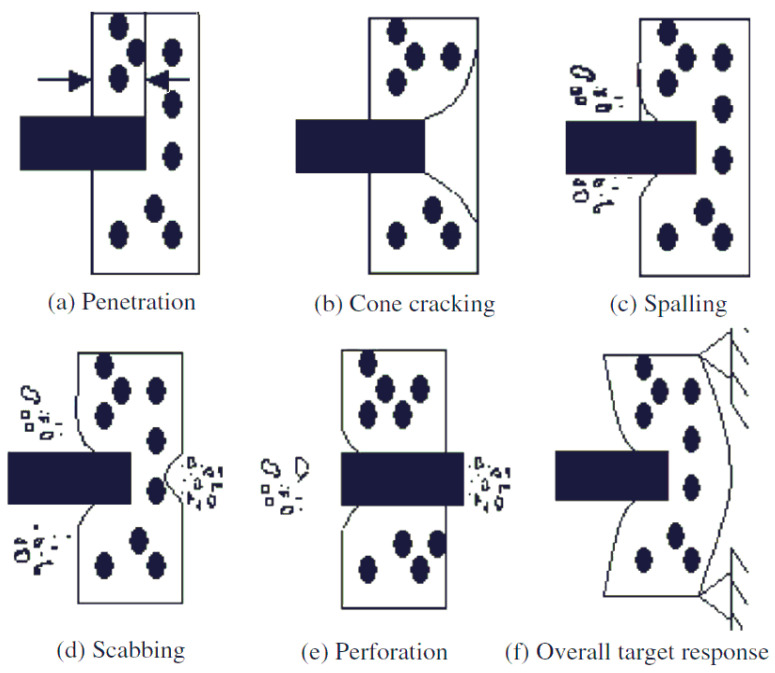
Identification of possible failure modes [15] into concrete target: (**a**) penetration into the target; (**b**) cone cracking and plugging due to the inertial stress wave; (**c**) spalling with the ejection of target material from the proximal face of the target; (**d**) radial cracking behind the initial wave front; (**e**) scabbing with the ejection of fragments from the distal face of the target; (**f**) potentially, perforation completely through the target.

**Figure 7 materials-14-00816-f007:**
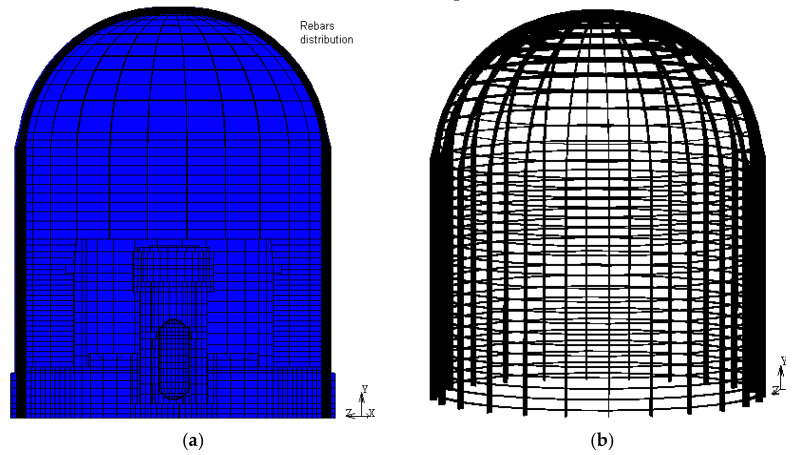
Vertical section of the CC model (**a**) and overview of the reinforcement arrangement (**b**).

**Figure 8 materials-14-00816-f008:**
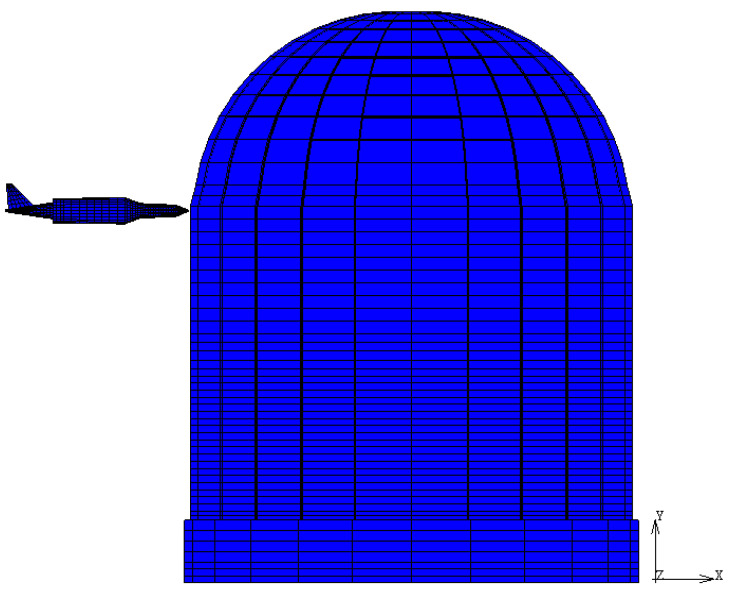
Horizontal aircraft impact configuration. Models represent the explicit interaction of the missile and target by means of a gap contact.

**Figure 9 materials-14-00816-f009:**
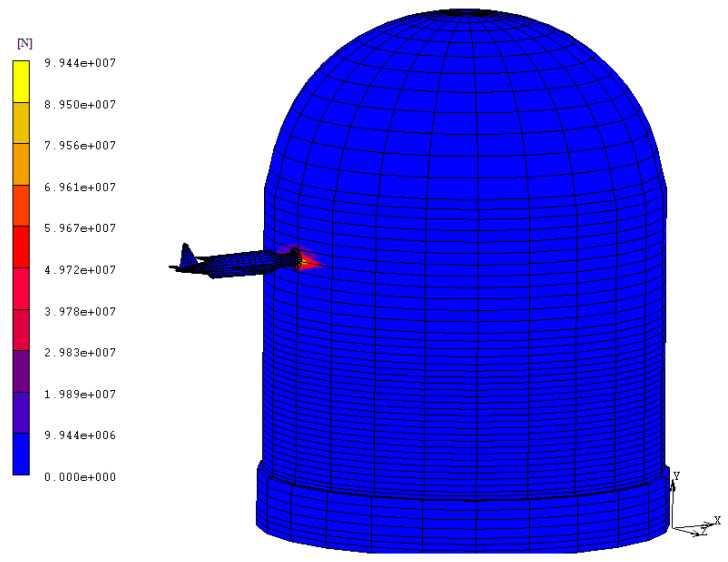
Impact force on target.

**Figure 10 materials-14-00816-f010:**
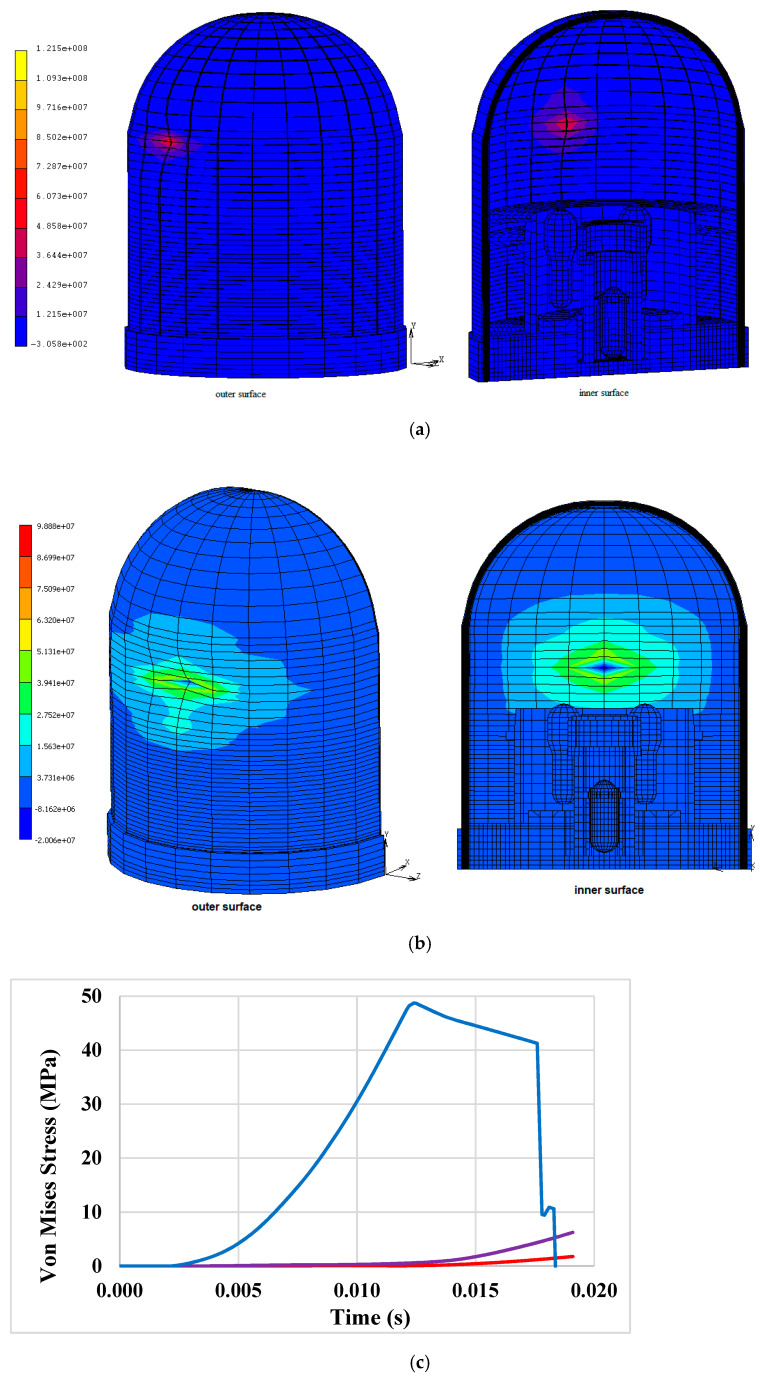
Von Mises stress (Pa) distribution at fuselage crashing for (**a**) unaged and (**b**) aged CC. The figure (**c**) shows the stress trends at several points located inside the aged CC: the blue line represents the behavior at the entry point, the violet the behavior at the reactor cavity pit, and the red curve the behavior at almost the opposite of the entry point.

**Figure 11 materials-14-00816-f011:**
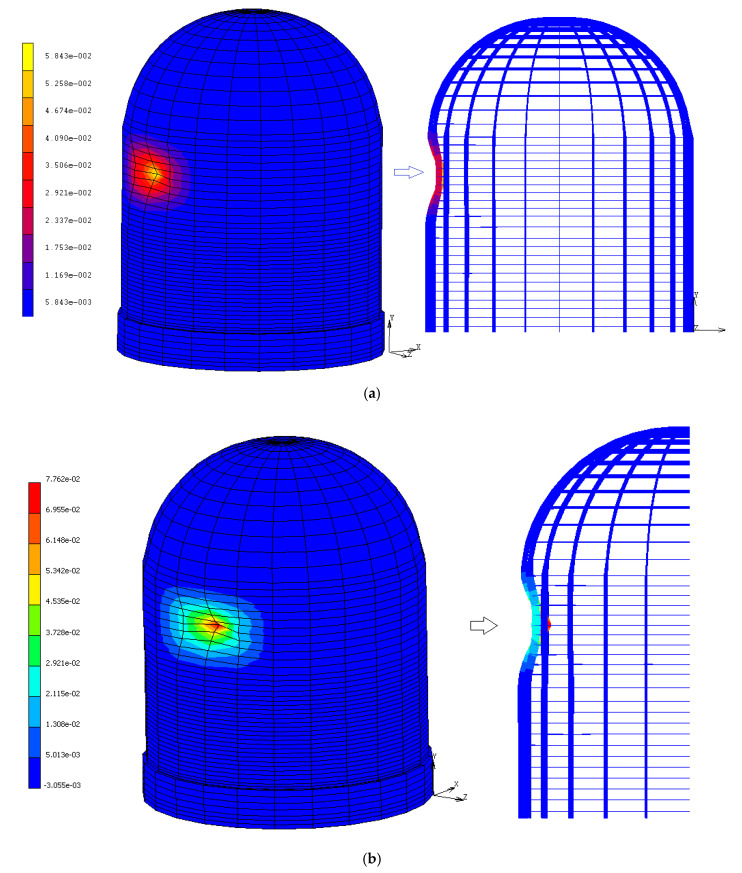
Penetration depth (m) for (**a**) unaged and (**b**) aged CC.

**Figure 12 materials-14-00816-f012:**
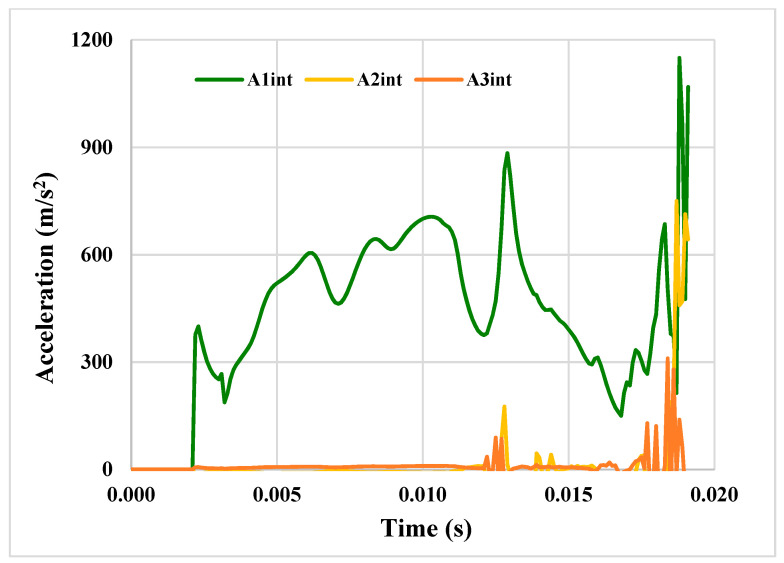
Acceleration through the aged containment structure.

**Table 1 materials-14-00816-t001:** Material properties of the rebar and studs.

	Rebar	Stud	Concrete
Density (kg/m^3^)	7850	7850	2400
Young’s Modulus (MPa)	210,000	200,000	180,000
Poisson Ratio (-)	0.3	0.3	0.2
Yield Stress (MPa)	375	345	48
Elongation to Fracture (%)	>14	12	

## Data Availability

The data presented in this study are available on request from the corresponding author

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
