# Peer review of "Aircraft Impact Effects on an Aged NPP"

_materials, 2021, doi:10.3390/ma14040816_

Round 1
Reviewer 1 Report
The paper concerns the modeling of deformation processes that occur during the dynamic interaction of a flying missile and a reinforced concrete target.
The essence of mathematical modeling is presented very superficially, only the verbal formulation of the problem is given and the MARC package options used are mentioned. The absence of a mathematical formulation of the problem makes it difficult to understand the essence of the mathematical model of the considered physical processes.
In particular, it is impossible to understand from the text which version of the model of elastoplastic deformation is used. It is impossible to assess how the physical properties of construction materials change during aging. The text does not provide information on how the processes of fragmentation of colliding bodies are modeled.
This raises the question of how the mechanical properties of the projectile and design materials are taken into account. Table 1 lists only the properties of the metal components of the structure. It does not say anything about the properties of concrete and missile. In addition, the tabular data refer to low-speed deformation processes.The author does not substantiate the possibility of their use in modeling high-speed processes where the limiting characteristics depend on the deformation rate.
These circumstances do not allow us to assess the reliability of the results obtained, although the words concerning the nature of the deformation interaction do not raise objections. The words about the very good agreement between the results of simulation and full-scale aircraft impact test are only descriptive.
Notes on the article design
Line 75-76 7m2 wide either the linear dimensions of the target, or its area
Fig 3b figure caption is incomprehensible, there is no dimension for the scale
Fig 3, 4 what moment in time do the pictures correspond to?
Fig 9 the upper graph is unreadable, there is no explanation to this sraph
Fig 10 a, b, Fig 11 what moment in time do the pictures correspond to?; scales have no dimensions
Fig 12 no explanation of what lines 1, 2, 3 mean
Reviewer 2 Report
Some images have a low resolution (they are not quite clear)
Reviewer 3 Report
In the present manuscript, the study addressed the numerical simulation of an aged NPPs subjected to an aircraft impact. The effects of impact velocity, direction and location were investigated together with the unfavourable conditions to be expected for the plant. Firstly, the numerical model was validated based on the results obtained from the experiments of Sugano et al. Through the results comparation between the aged an unaged NPPs, the author showed that ageing increases the plant propensity to suffer extensive, even penetration. Meanwhile, the research indicated that the damage at the impact area confirmed to be dependent on the type of aircraft and target wall thickness has a great important meaning not only in the academic but also in the practical engineering. I recommend to accept it after major revisions, the following aspects should be paid attentions:
(1) In the abstract, line 12, the expression that “This study addresses the numerical simulation of an aged NPP subjected to a large commercial aircraft impact”. But in the manuscript only the F-4 fighter was used, which is not a large commercial aircraft.
(2) Line 166, author expressed the two-strike locations have been selected:1) just above the equipment hatch; 2) on the dome. However, the equipment hatch location was not found in FEM model. Then in the simulation discussion section, there were no relevant results or explanation.
(3) The manuscript does not describe the material behaviour of the concrete and the steel materials. The study is based on a highly nonlinear analysis and the deformations would be strain rate dependent. No explanation has been given regarding modelling the behaviour of material, either for steel or concrete. Table 1 is insufficient to describe the behaviour of concrete for the problem studied.
(4) The author should describe the difference between aged and unaged concrete of containment material behavior.
(5) The containment model mesh gird is relevant large (4.5 m element size estimated). In previous published papers(Zou & Sui et al. The Structural Design of Tall and Special Buildings, 2019, 28(16)), the mesh gird adopted should be fine enough. Please explain the reason and the reasonableness of the results based on the mesh size.
Round 2
Reviewer 3 Report
Accept